# Herbivory in *Myrtillocactus geometrizans* (Cactaceae): Do Parasitoids Provide Indirect Defense or a Direct Advantage?

**DOI:** 10.3390/plants12010047

**Published:** 2022-12-22

**Authors:** Alicia Callejas-Chavero, Diana Guadalupe Martínez-Hernández, Carlos Fabian Vargas-Mendoza, Arturo Flores-Martínez

**Affiliations:** 1Laboratorio de Ecología Vegetal, Departamento de Botánica, Escuela Nacional de Ciencias Biológicas, Instituto Politécnico Nacional, Mexico City 11340, Mexico; 2Laboratorio de Variación Biológica y Evolución, Departamento de Zoología, Escuela Nacional de Ciencias Biológicas, Instituto Politécnico Nacional, Mexico City 11340, Mexico

**Keywords:** *garambullo*, soft scale, parasitoids and ants

## Abstract

Plants respond to herbivory in diverse, complex ways, ranging from avoidance or tolerance to indirect defense mechanisms such as attracting natural enemies of herbivores, i.e., parasitoids or predators, to strengthen their defense. Defense provided by parasitoids to cultivated plants is well documented and is used in biological control programs. However, its effectiveness on wild plants under natural conditions has been little studied. Such is the case of the cactus *Myrtilllocactus geometrizans* (known in Mexico as *garambullo*), which is consumed by the soft-scale insect *Toumeyella martinezae* (herbivore) which, in turn, is host to the parasitoid wasp *Mexidalgus toumeyellus*, and mutualist with the ant *Liometopum apiculatum,* that tenders and protects it. This study explores the role of the parasitoid as an indirect defense, by examining its effect on both the herbivore and the plant, and how this interaction is affected by the presence of the mutualistic ant. We found that scales adversely affect the cactus’ growth, flower, and fruit production, as well as its progeny’s performance, as seedlings from scale-infested *garambullo* plants were shorter, and it also favors the presence of fungus (sooty mold). The parasitoid responded positively to herbivore abundance, but the presence of ants reduced the intensity of parasitism. Our results show that parasitoids can function as an indirect defense, but their effectiveness is reduced by the presence of the herbivore’s mutualistic ant.

## 1. Introduction

Interactions between organisms in nature usually form a complex web encompassing various trophic levels [1]. Until recently, most studies on plant–insect interactions used to focus on direct relationships (plant–herbivore, plant–pollinator, or plant–disperser), overlooking other interactions between those these species. It is now recognized that, to gain a more comprehensive picture of plant–insect interactions and better understand their outcomes, lower and higher trophic levels (i.e., predators, parasitoids, hyperparasitoids, and decomposers) [2,3,4], as well as competitors [5,6,7,8], must be included.

Plant responses to herbivory can be complex and includes traits that allow them to escape, defend, or tolerate the herbivores attack [9]. Defense induced by herbivores can be direct, such as the production of chemical compounds (e.g., saponins, tannins, and phenols) that affect the herbivore’s physiology or behavior [10,11]. Indirect defenses include mechanisms that ameliorate the impact of herbivores by attracting [12] or providing rewards or shelter to their natural enemies (predators, parasitoids, parasites, or pathogens), thus favoring their permanence on the plant and attack to herbivores [10,11,13]. In this scenario, plants are benefitted by reducing the adverse consequences of herbivores [14]. This indirect defense is flexible and can be affected by a range of biotic and abiotic factors [15].

That plants attract natural enemies of their herbivores by releasing herbivore-induced volatile compounds (HIPV) is well documented [16]. In parasitoid-mediated indirect defense mechanisms, parasitoids are attracted by volatile compounds produced by either the plant (tri-trophic interaction) [17,18,19] or the herbivore (bi-trophic interaction) [20,21,22,23]. Those compounds can also repel or inhibit the development of other herbivores [24,25].

This mechanism has been widely documented in crop plants such as maize, wheat, coffee, and cotton [11,19,26,27,28,29]. Little is known about these mechanisms in other plant groups, such as Cactaceae, although the presence of by-products or substance mixtures that could attract natural enemies of herbivores has been recorded. For instance, alkanes that contribute to the production of volatiles which, in addition to repelling herbivores, attract their natural enemies, have been found in the genera *Opuntia*, *Cylindropuntia*, *Grusonia*, and *Consolea* [25]. *Opuntia ficus-indica* plants attacked by the moth *Cactoblastis cactorum* have been reported to emit volatile compounds that attract the parasitoids *Microplitis croceipes* and *Apanteles opuntiarum* [30,31]. *O. ficus-indica* plants attacked by the red mealybug *Dactylopius coccus*, the prickly pear cochineal *D. opuntiae*, and the herbivore *Ceratitis capitata* have been found to produce kairomones [32,33,34,35].

Céspedes [36] pointed out that the high resistance of *Myrtillocactus geometrizans* to insect attack could be explained by the presence of triterpenes and sterols, compounds that have also been isolated from other cacti species such as *Lamaireocerus chichipe*, *Myrtillocactus cochal*, *M. schenckii*, and *M. eichlamii*. Although these compounds have shown positive effects in controlling common pests such as *Spodoptera frugiperda* and *Tenebrio molitor* in laboratory experiments, it is not known whether they have similar effects on natural herbivores of *M. geometrizans* or contribute to attracting their natural enemies (predators and parasitoids) [37].

Parasitoid characteristics, such as their high specificity, remarkable ability to locate host individuals, adaptation to changes in environmental conditions, synchronization with the host’s life cycle, ability to survive (diapause, dormancy) periods when the host is absent, density-dependent mechanisms, and high reproductive success, make them a potentially important defense for plants [32,33,38,39,40,41]. Aguirre [35] reported that the parasitoids *Anagyrus cachamai* and *A. lapachosus* (Hymenoptera: Encyrtidae) reduced the damage caused by the herbivore *Hypogeococcus* sp. (Hemiptera: Pseudococcidae) on *Hylocereus undatus* and contributed to maintaining cacti diversity in the area.

The range of interactions involved in indirect defense, as well as the associated ecological and evolutionary processes, have not yet been clearly elucidated [31,32,33]. Most studies on plant-parasitoid interactions have been carried out on plant crop species under laboratory conditions, with results indicating a high efficiency of parasitoids in controlling herbivores and a significant reduction in herbivory [41,42,43]. However, results are very likely to be different under natural conditions, since plants can interact simultaneously with other species (herbivores, pathogens, mutualists, etc.), with positive or adverse effects of varying intensity at the individual and population level [44]. This makes it difficult to elucidate the herbivore’s effect and importance, and to predict the outcome of the interaction (or at least the plant’s response). It is also difficult to identify the plant trait (for example, volatile compound emitted) that attracts parasitoids and assign it an active defensive role [44,45,46].

The interaction between the cactus *Myrtillocactus geometrizans* (Mart. ex Pfeiff) Console, 1897 (Cactaceae) and its herbivore, the soft scale Toumeyella martinezae Kondo y Gonzalez, 2014 (Hemiptera:Coccidae), very clearly shows the importance of considering other inter-specific interactions that occur simultaneously. In addition to interacting with the *garambullo*, the scale *T. martinezae* maintains a competitive relationship with another scale species (*Opuntiaspis philococcus*) Cockerel, 1893 (Hemiptera: Diaspididae) [7], and is attacked by at least two parasitoid species that hide it from reaching the adult stage [47,48], and maintains a mutualistic relationship with the ant *Liometopum apiculatum* Mayr, 1870 (Hymenoptera: Formicidae), which takes care of it in return for a sugary liquid (honeydew) that it utilizes as food. In addition, when this scale is highly abundant, the excess honeydew deposited on the plant stems promotes the growth of a fungus that kills the plant in a short time [49].

The mutualistic interaction between plants and ants has been well documented [50,51]. However, when the ants also interact with other species, such as the plant’s phytophagous herbivores, modifying their abundance, the net effect on the plant may be different [47,52].

This study aimed to evaluate the likely role of the specialist parasitoid *Mexidalgus toumeyellus* Kondo y Gonzalez, 2014 (Hymenoptera: Aphelinidae) as an indirect plant defense, by examining its effect on both the herbivore *T. martinezae* and the cactus *M. geometrizans*, and how this interaction is affected by the presence of the herbivore’s mutualistic ant *L. apiculatum*. We hypothesized that, if this parasitoid acts as an effective indirect plant defense against the soft scale, it will respond to variations in the scale abundance and will reduce herbivory on the plant. In addition, the parasitoid effect on the scale population would be modified by the presence of the mutualistic ant, given the protection it provides to the scale.

## 2. Results

### 2.1. Effect of the Soft Scale T. martinezae on M. geometrizans

The phytophage effect on the plant’s fitness was determined by comparing several attributes between control (scale-free) plants and plants with a soft scale. The effect of the phytophage density on these attributes was also evaluated.

We found that scale-free *garambullo* branches grew twice as much, on average, then than those infested by scales (4 cm vs. 2 cm, respectively: F_1,54_ = 29.006, *p* = 0.004). The relationship between the growth of infested branches and phytophage density showed a significant negative slope, indicating a reduced growth when the phytophage is more abundant (Figure 1).

The reproductive success of *M. geometrizans* in the presence or absence of the scale was analyzed by means of a nested GLM (branches nested within individual plants). We found that scale-free *garambullos* produced significantly more buds (F_6,54_ = 3.255, *p* = 0.008) and flowers (F_6,54_ = 2.865, *p* = 0.017) than *garambullos* infested with scales (Figure 2a,b). However, the number of mature fruits was not significantly different (F_6,54_ = 2.138, *p* = 0.064) between infested and non-infested *garambullos* (Figure 2c)

As to the effect on the next generation, we found that the number of seeds germinated along the 44 weeks did not differ between *garambullos* with and without scales (W = 268.5; z = 1.107; *p* = 0.264). Although fruit size did not differ significantly between conditions (Figure 2d), seed number and size were greater in scale-free than in scale-bearing *garambullos* (Figure 2e). However, seedling size was significantly different (U = 38.5, *p* < 0.001), with seedlings from scale-free plants being larger than those from scale-bearing plants (Figure 2f).

Scale-infested plants were significantly more vulnerable than scale-free plants. The area affected by fungi (which may indicate the plant’s defense response) was greater (15%) in the former than in the latter (0.1%) (F_1,54_ = 54.606, *p* = 0.001). The trend line in Figure 3 shows that the area affected by fungi increases with the number of insects, thus revealing a greater vulnerability to fungal infestation.

### 2.2. Infestation Dynamics and Response to Scale Density

The proportion of stage-2 and stage-3 scales that were parasitized varied over time (F_1,134_ = 25.71; *p* < 0.001; F_1,134_ = 21.62, *p* < 0.0001, for time and time^2^, respectively), and exhibited different dynamics depending on the presence or absence of ants (Figure 4). During the first two months of the study, the proportion of parasitized scales was similar (about 15%) in *garambullo* plants accessible to ants and in those where ants were excluded. However, the proportion of parasitized scales decreased significantly in the presence of ants, reaching an average of about 10%. In contrast, the percentage of parasitized scales in the population not tended by ants continued to increase until an average of 25% (Figure 5).

The abundance of parasitized scales was significantly related to the density of stage-2 and stage-3 scales in the population (F_5,134_ = 69.73, *p* < 0.001), the more scales available, the larger the number of parasitized scales. However, this response was not linear; the number of parasitized scales in plants accessible to ants leveled off at intermediate densities (approx. 1000 individuals). In contrast, the number of parasitized scales in plants where ants were excluded increased significantly faster with scale density (F_1,134_ = 4.734, *p* = 0.0313), but this number decreased markedly at higher scale densities (F_1,134_ = 4.971, *p* = 0.0274), likely due to the excess honeydew present, which reduced their quality as prey.

## 3. Discussion

Herbivory is perhaps the most common interaction in nature. However, its role in the growth and distribution of plant populations is still under discussion [53], especially in long-lived species inhabiting stressful environments [54] such as cacti [37]. The few studies available on insect herbivory in cacti have focused on commercial species growing in relatively homogeneous culture conditions [55,56], even though herbivory intensity has been shown to vary depending on environmental conditions [57]. Other studies have listed the herbivore species associated with various cacti genera but have not evaluated their effect on plant performance [58].

Soft scales are important pests for various plant species, including cacti, due to their direct (necrosis due to the piercing by the mouthparts and extraction of resources) and indirect (pathogen transmission, surface contamination, and plant weakening) effects [59]. Our results show that the presence of the scale *Toumeyella martinezae* significantly reduces the *Myrtillocactus geometrizans* fitness by affecting its growth, reproduction, and its progeny’s performance, and that these effects vary with the abundance of herbivores, as has been reported in other systems [54,60,61,62,63]. Although *M. geometrizans* shrubs exhibit some resistance and tolerance to herbivory at low scale densities, when *T. martinezae* is highly abundant its adverse effects become evident. Céspedes [36] pointed out that the *garambullo*’s resistance to insect attack may be associated with the presence of insect growth regulators such as triterpenes and sterols; this hypothesis remains to be tested.

Few studies have evaluated the effect of herbivores on the plant’s vigor or vulnerability to diseases or to the attack of other herbivores. For *garambullo* plants, the presence and abundance of scales was also related to more necrotic tissue and the presence of fungus (sooty mold) covering its photosynthetic stems, thus reducing its ability to photosynthesize and grow.

Despite the wide distribution of *M. geometrizans* in the arid zones of central Mexico [64,65], its interaction with the scale *T. martinezae* and the parasitoid *M. toumeyellus* has only been reported for a particular location in central Mexico, where this study was conducted. In fact, no other host species for this scale and the parasitoid have been reported to date [37,47,49,66]. Micro-endemisms in Hemiptera such as *Nysius wekiuicola* have been documented in locations with very particular environments distinctive environmental conditions [67]. These highly specific relationships pose a strong selection pressure especially on the phytophage. *M. geometrizans* is the sole resource for the scale *T. martinezae*, and any response of the plant towards the scale will have important consequences on its adaptation and defense. Similarly, the abundance and availability of scales in the elective stage determine the oviposition success of *M. toumeyellus*. This suggests that natural selection should favor optimization mechanisms in the parasitoid, promoting its host specialization [44,68].

In addition to the main traits (visual, olfactory, vibratory stimuli, and chemical signaling) utilized by parasitoids for locating and ovipositing on a host [31] the amount of HIPV has been documented to depend on herbivore density. When the herbivore density increases, damage to the plant also increases, which increases the emission of volatile compounds that parasitoids detect as a clue for oviposition [69]. However, under natural conditions, host-seeking parasitoids are exposed to the variety of volatiles emitted by all non-host plants growing in the surrounding vegetation, which can undermine the parasitoids’ olfactory orientation capability [70,71,72].

Although the compounds acting as attractants for parasitoids in the *garambullo*-scale system have not been identified, the signals perceived by *M. toumeyellus* are clearly sufficient to locate the scattered populations of *T. martinezae* present in the area. Although *M. geometrizans* is an important component of the local vegetation, their density is only 48.5 individuals per hectare, less than 50% of the individuals have scales (personal obs./unpublished data), and plants infested with scales are, in most cases, several tens of meters apart. The parasitoids’ effectiveness at locating scale populations may be related to the high scale density on each plant, which would result in larger emissions of volatile compounds [73]. The ant plays an important role in this process, as it promotes the rapid growth of scale populations, making them more attractive to parasitoids; it is well documented that parasitoids respond to site quality, staying longer in sites where more potential hosts are available [74]. Nevertheless, some studies have observed that parasitoids do not respond to [75] or even show a negative relationship with [76,77] prey density, avoiding ovipositing in populations with high host density. *Mexidalgus toumeyellus* responded positively to scale density, as the number of parasitized scales increased with scale density, although this relationship was not linear. The proportion of parasitized scales is usually lower than 15% under natural conditions (with ants), which could be due to the females’ age, the fact that parasitoids become satiated, or because they do not perceive it as a safe site [78].

The presence of mutualistic ants adds to the complexity of the plant–scale–parasitoid system, and their direct and indirect interactions with parasitoids and plants make it even more difficult to identify their net effects. Although the plant–ant mutualism is well documented [50,51,79,80] in our study system, the ant seems to have a net adverse effect on the plant as it favors its herbivore.

There are several examples of mutualistic interactions between ants and other insect species where a reward is obtained in return for the defense ants provide against enemies [81,82]; for example, the interaction between ants and honeydew-producing insects [83]. In these cases, ants tend to their mutualists by relocating, grooming, and defending them from natural enemies [84,85]. In our study system, the ant *L. apiculatum* performs all these functions with the scale and our results show that this also reduced the incidence of parasitism. This may be due to the ants’ continuous patrolling over the scales, which prevents the parasitoids from landing and ovipositing on them, as has been reported in other systems [86]. The incidence of parasitism is reduced in almost 50% of protected scales; although substantial, this is not enough to stop the growth of the scale populations and their consequent direct and indirect adverse effects on the plant.

Although the ants could favor the plant by providing defense against other herbivores [79,80], the care they provide to the scales has indirect effects that are adverse for the plant and the other participants of this interaction. The excess honeydew in large scale populations promotes the growth of fungi (sooty mold), a phenomenon that has been widely documented in other plant species [87,88]. The sooty mold covers the plant stems, hampering photosynthesis, reducing the plant’s vigor, and, when highly abundant, the *garambullo* plant dies in a few years [47]. This is the main cause of mortality in adult *M. geometrizans* in the study area. Paradoxically, the death of the plant also entails the end of the scale population and its interactions.

The shrub–scale–parasitoid association seems to follow a metapopulation dynamics in which the colonization of new plant individuals by the scale (with/without the help of ants), the location of parasitoids, and the promotion of growth by the ant, determine the dynamics and growth rate of all the system participants.

When ants are excluded, the parasitoid effect changes substantially and the proportion of parasitized scales increases rapidly with scale density, reaching values higher than 30% before decreasing substantially. This decrease may be due to the excess honeydew that acts as a self-contaminant that causes the death of scales [48,89,90] reduces their quality as hosts [88,91] or inhibits parasitoid oviposition if honeydew contains plant defense compounds [88,92,93]. Other likely explanations for this negative relationship with density include the likely presence of hyperparasitoids or other predators that could be attracted by the abundance of herbivores and would affect the parasitoid emergence [4].

As in other systems [79,80] where ants (particularly *L. apiculatum*) are involved, the relationship with the scale seems to be a facultative mutualism. In the field, ants are often observed patrolling or visiting other plant species (*Opuntia*, Agavaceae, Rhamnaceae, *Schinus molle*, etc.), but they provide pay substantial attention to the scales on *garambullo* plants. We did not find scale populations that were not tended by ants. We still must determine whether the scales are completely dependent on the ants, or the ants are highly efficient at locating all the scale populations and exploiting them to obtain honeydew.

In summary and trying to classify the interactions involved in our study system, in particular the relationship between plants and parasitoids, our results show that parasitoids can play an important role as indirect defense as they significantly reduced scale density and ameliorated the herbivore’s adverse effect. Thus, the relationship between the plant and the parasitoid would be a mutualism obligated for the parasitoid, given its specificity. However, the effectiveness of this defense mechanism changes when the ant is present, as the rate of parasitism decreases significantly. This implies that the parasitoid continues to benefit by finding scales to oviposit, but the plant receives only a marginal benefit, making the interaction closer to commensalism, with little collateral direct advantage to the plant. Thus, the phrase “my enemy’s enemy is my friend”, often used to describe the mutualism between plants and natural enemies, does not always hold.

Our study showed the importance of adopting a multitrophic approach in studies of parasitoid-mediated indirect defense under natural conditions, as this allows a better evaluation of the importance of interactions and better identification of mechanisms with evolutionary potential and patterns that can be used for herbivore control.

## 4. Methods

### 4.1. Study System

*Myrtilllocactus geometrizans* (Mart. ex Pfeiff.) Console, 1897 (Cactaceae), is a cactus endemic to Mexico, where it is common in the arid zones of the central part of the country and important from an ecological and social point of view (Figure 6a). This cactus establishes direct and indirect relationships with numerous arthropod species. Two phytophagous hemipteran species, known as scales, have been identified to date, the armed scale, *Opuntiaspis philococcus* Cockerel, 1893 (Hemiptera:Diaspididae) and the soft scale *Toumeyella martinezae* Kondo y Gonzalez, 2014 (Hemiptera: Coccidae).

*T. martinezae* has two generations during the year, each lasting for approximately five months. The first generation starts at the onset of the rainy season (late May, early June) and the second in late October, early November. Our study only followed-up the June to November generation. *T. martinezae* forms a mutualistic relationship with the *escamolera* ant, *Liometopum apiculatum* Mayr, 1870 (Hymenoptera: Formicidaea) (Figure 6b); the ant grooms and relocates the scales to places favorable for their establishment, and protects them from natural enemies, in exchange for a carbohydrate-rich liquid (honeydew) that it utilizes as food. The soft scale is unable to produce honeydew, and no association with ants has been reported. When in their first development stage, both scale species move around and colonize young branches of the plant, where they eventually settle down and remain sessile until completing their life cycle. The two scale species can occur simultaneously in *garambullo* plants and are, in turn, used as hosts by various parasitoid species [49,94] particularly by the parasitoid *Mexidalgus toumeyellus* Myartseva et al., 2014 (Hymenoptera: Aphelinidae), which has been identified as a specialist for *T. martinezae* (Figure 6c). When soft scale populations become very large, the honeydew secreted and not used by ants facilitates the growth of a fungus (*Fumagospora* sp.) that causes the death of branches and, in some cases, of the entire plant (Figure 6d).

### 4.2. Field Work

The study was carried out in a xerophytic shrubland near the Zequetejé town, municipality of Huichapan. It is located to the west of the State of Hidalgo, Mexico, between parallels 20°22′24″ N and 99°38′56″ W, to an altitude of 2100 m above sea level. Its climate is very dry, with summer precipitation (BW i′(w)g) and an annual temperature of 16 °C; the average annual rainfall is 437 mm, with a rainy season from May to September. The soils are of volcanic origin, well-drained, and rich in organic matter [48]. To evaluate the effect of the scale insects on *M. geometrizans*, 20 individual plants that were at least 2.5 m tall were randomly selected, ten plants were bearing the soft scale *T. martinezae* and the other ten were free of scales. Four branches were selected on each plant. Ants were removed from two branches on each of the scale-bearing plants and their later access was prevented by means of a ring of solid, unscented petroleum jelly at the base of the branch. Free access of ants was allowed in the other two branches. Four branches were selected in each of the scale-free plants. The length of each branch was measured, and the presence of fungi damage was recorded.

To evaluate the effect of the phytophage on the growth of *garambullo* plants, the branch length growth of scale-free and scale-bearing plants was compared by means of a nested generalized linear model (GLM). Branch growth was the response variable, plant condition was the fixed effect factor, and the branches nested in each plant was a random effect factor. This analysis was performed in SPSS v25 [95].

The same design was used to evaluate how susceptible or vulnerable the plants are to fungal infestation when attacked by scales. In this case, the percent area covered by fungi was the response variable; this percentage data was arc sine transformed prior to analysis. A GLM was fitted with attack condition as the fixed effect factor and the branches nested in each plant was a random effect factor (four branches per plant).

The reproductive success of plants with or without scales was evaluated in terms of the number of (1) flower buds, (2) flowers, and (3) fruits. Since these response variables were measured in the same four branches on the same plants, and there was a strong correlation between them, the data were analyzed using a nested MAMOVA. Plant condition was the fixed effect factor and the branches nested in each plant were a random effect factor. These analyses were performed in SPSS v25 [95].

To evaluate the effect of scale attack on the next plant generation, that is, on germination and seedling establishment, a batch of 1000 seeds produced by either scale-free (control) or scale-infected plants were selected. The 2000 seeds were monitored for 44 weeks and their time to germination was individually recorded. The size of 94 seedlings from scale-free mother plants and 82 seedlings from scale-infested plants was measured 30 days after emergence. A paired Wilcoxon test was used to analyze the germination data (the germination week was the pared criterion between the attacked and non-attacked plants); seedling size data were compared using a non-parametric Mann–Whitney test. These analyses were performed in SPSS v25 [95].

To analyze the dynamics of parasitoid infestation and its response to scale density, the density of scales on branches of 12 randomly selected shrubs was recorded during June–November 2016. All the scales present in a 5-cm-wide ring placed on each branch were collected, counted, and examined under a stereoscopic microscope to determine their development stage and whether they were infested by the parasitoid. Ants were excluded from half of the branches, as described above.

GLMs [96] were used to examine the dynamics of parasitoid infestation and the relationship between parasitized scales and scale abundance. For the first case, the proportion of parasitized scales was the response variable, linear, and quadratic components of time were continuous predictors, treatment (with or without ants) was a fixed-effect factor, and a binomial error distribution and a logit link function were used. To examine the relationship between parasitized scales and the abundance of stage-2 and stage-3 scales (the stages in which the larvae are parasitized), a Poisson error distribution and a log-linear link function were used. The model included the (linear and quadratic) abundance of scales as a continuous factor and the presence/absence of ants as a fixed-effect factor. When needed, the analyses were corrected for overdispersion by rescaling the residual deviance with the error degrees of freedom [96].

## Figures and Tables

**Figure 1 plants-12-00047-f001:**
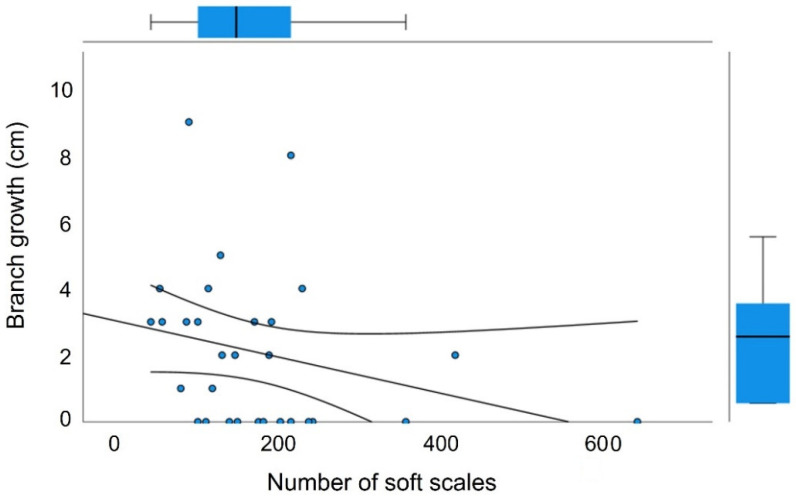
Relationship between the number of herbivores (*Toumeyella martinezae*) and branch growth, showing a decreasing trend. Boxplots along the axes’ margins show the dispersion of variable data; the trend line and corresponding 95% confidence bands are also shown.

**Figure 2 plants-12-00047-f002:**
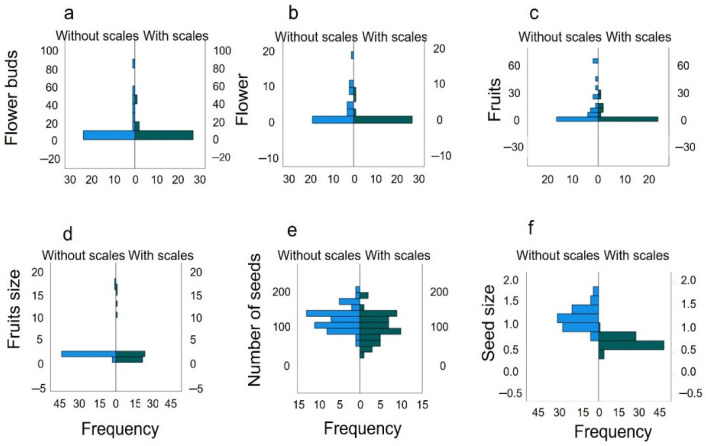
Comparison of the frequency distributions of (**a**) flower buds, (**b**) flowers, (**c**) fruits, (**d**) fruits size, (**e**) number of seeds, and (**f**) seed size in control plants (in blue) and plants infested by the phytophagous *Toumeyella martinezae* (green).

**Figure 3 plants-12-00047-f003:**
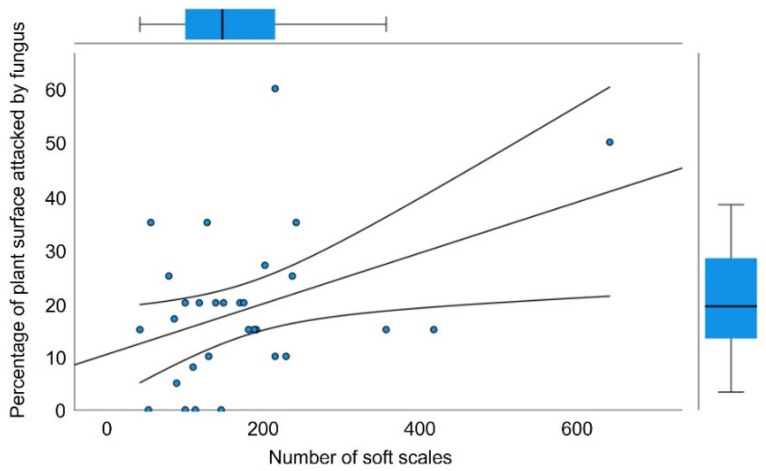
Relationship between the number of herbivores (*Toumeyella martinezae*) and percent area affected by fungi, showing an increasing trend. Boxplots along the axes’ margins show the dispersion of variable data; the trend line and corresponding 95% confidence bands are also shown.

**Figure 4 plants-12-00047-f004:**
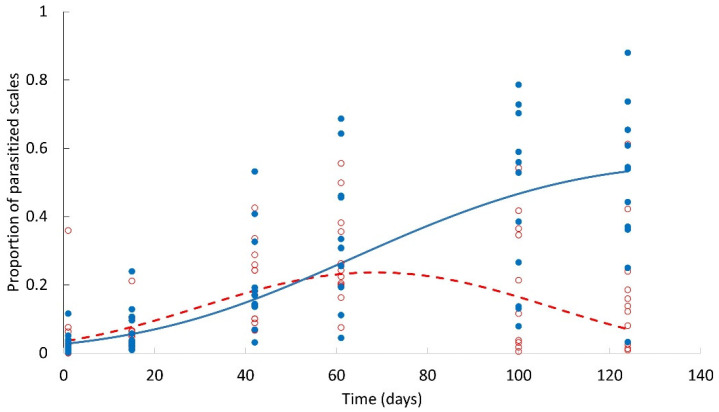
Proportion of scales of stages 2 and 3 parasitized by *Mexidalgus toumeyellus* throughout the life cycle of *Toumeyella martinezae*, in presence and absence of its mutualistic ant *Liometopum apiculatum*. Blue solid circles correspond to the treatment without ants and empty red circles to the treatment with ants. The solid blue line corresponds to the model adjusted for the treatment without ants and the dotted red line with the presence of ants.

**Figure 5 plants-12-00047-f005:**
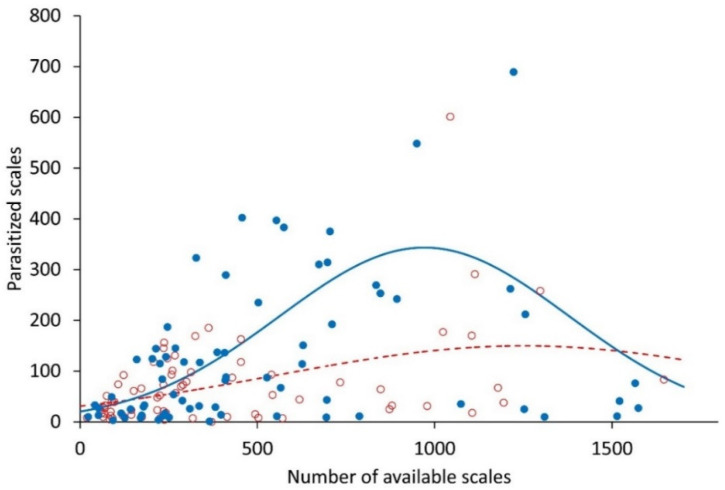
Number of parasitized scales depending on the available scales of stages 2 and 3 in the presence and absence of their mutualistic ant *Liometopum apiculatum*. Blue circles correspond to the treatment without ants and empty red circles to the treatment with ants. The solid blue line corresponds to the model adjusted for the treatment without ants and the dotted red line at treatment with ants.

**Figure 6 plants-12-00047-f006:**
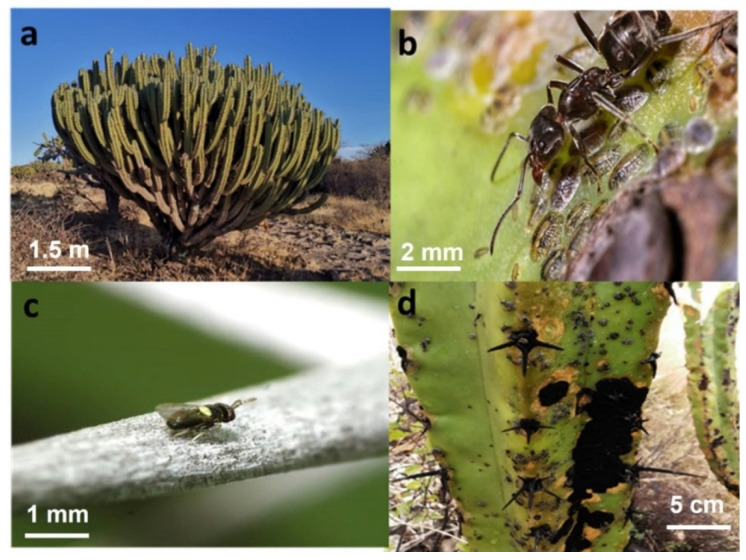
(**a**) *Myrtillocactus geometrizans*, (**b**) *Liometopum apiculatum* tending to *Toumeyella martinezae,* (**c**) *Mexidalgus toumeyellus* on *M. geometrizans* thorns and (**d**) branches of *M. geometrizans* showing damage, lignification, and presence of fungi due to the presence of *T. martinezae*. Images: Jesús Luna and Arturo Flores Callejas.

## Data Availability

Data will be made available on request.

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
