# Peer review of "Herbivory in *Myrtillocactus geometrizans* (Cactaceae): Do Parasitoids Provide Indirect Defense or a Direct Advantage?"

_plants, 2022, doi:10.3390/plants12010047_

Round 1

Reviewer 1 Report

The authors have made a valuable contribution to science by producing paper documenting their well-designed, field-based study of a multitrophic, complex system involving the interaction of scale insects, parasitoid wasps and ants on garambullo plants under natural conditions. The aims of the study are clearly stated, statistical analyses appropriate and the figures included are effective and informative. This paper will be of interest to a diverse range of researchers including plant ecologists and entomologists.

Review comments/suggestions

Title: Herbivory in Myrtillocactus geometrizans (Cactacea): Are Parasitoids an Indirect Defense or Collateral Benefit? “Collateral benefit” is a legal term rather than a scientific description of an association.

Suggested changes:

Herbivory in Myrtillocactus geometrizans (Cactacea): Do Parasitoids provide Indirect Defense or a Direct Advantage?

Summary

Page 1, Line 10

Plants respond to herbivory in diverse, complex manners ways, ranging from ranging from avoidance or tolerating……..

Page 1, Line 11

natural enemies of herbivores, such as parasi-………

Page 1, Line 15 – To let those readers who are not familiar with these insects know exactly what they are:

…..the soft scale insect, Toumeyella mar-….

Page 1, Line 16

……parasitoid wasp, Mexidalgus toumeyellus

Introduction

Page 1, Line 28

……various trophic levels [1].

Page 1, Line 30

…..between those these species.

Page 1, Line 37

……or behavioral [10,11].

Page 2, Line 67

…contribute to attracting their natural enemies (predators….

Page 2, Line 69

Parasitoids characteristics such as their………..

Page 2, Line 73

………plants [32,33,38-41]. Author/s names missing? reported that the parasitoids…….

Starting Page 2, last paragraph lines 89 to 103, are a different (sans serif) typeface.

Results

Page 3, Line 117

………twice as much, on average, then than……..

Page 4, Figure 2. Axis labels -  Frecuency should be Frequency – as correctly stated in the Figure legend.

Discussion

Page 7, Paragraph 2, lines 203 to 207, are a different (sans serif) typeface.

Page 7, Line 213

….documented in locations with very particular environments distinctive environmental conditions [68].

Page 7, Line 244

…under natural conditions (with ants ’s), which could be due to the……

Page 7, Lines 276 to 284 are a different (sans serif) typeface.

Page 7, Line 288

but they provide pay substantial attention to the scales on…….

Page 7, Line 300

little collateral direct benefit (or advantage)…..

Methods

 Page 9, Line 312

….phytophagous hemipteran species, known as scales…..

Acknowledgements are also in different typeface and large point size.

Author Response

We have meticulously reviewed the manuscript and have given answers to the comments stressed by you. In this cover letter we are presenting the point-by-point answers.

ANSWER FO OTHER NOTES

Its main notes are:

  • Use full name (plants and Insects) with author, order, family when first mentioned in text.

ANSWER: We added this information in text.

Pages 3 and 4, lines 102, 103, 104, 108, 110, 119

  • Reader usually wants to read how the authors conducted the study then reads the results. Material and methods section should be before results, also change figure numbers accordingly. Unless this is the style for section: Plant Ecology.

ANSWER: We follow the editorial rules of Plants in this regard.

  • Add more details on site of study with information on weather condition during this evaluation.

ANSWER: In page 14, lines 373-379, we add more general environmental information of study site

Review 1

Title: Herbivory in Myrtillocactus geometrizans (Cactacea): Are Parasitoids an Indirect Defense or Collateral Benefit? “Collateral benefit” is a legal term rather than a scientific description of an association.

Suggested changes:

Herbivory in Myrtillocactus geometrizans (Cactaceae): Do Parasitoids provide Indirect Defense or a Direct Advantage?

ANSWER: We done the changes of Title as the reviewer suggests

Summary

Page 1, Line 10

Plants respond to herbivory in diverse, complex manners ways, ranging from ranging from avoidance or tolerating……..

ANSWER: Changes were done, Page 1 line 17

Page 1, Line 11

natural enemies of herbivores, such as parasi-………

ANSWER: Change was done, Page 1 line 19

Page 1, Line 15 – To let those readers who are not familiar with these insects know exactly what they are:

…..the soft scale insect, Toumeyella mar-….

ANSWER: Page 1 line 23, this line changed: ….. garambullo), which is consumed by the soft scale insect Toumeyella martinezae (herbivore)….

Page 1, Line 16

……parasitoid wasp, Mexidalgus toumeyellus

ANSWER: Change was done, Page 1 line 24

Introduction

Page 1, Line 28

……various trophic levels [1].

ANSWER: Change was done, Page 2 line 41

Page 1, Line 30

…..between those these species.

ANSWER: Change was done, Page 2 line 43

Page 1, Line 37

……or behavioral [10,11].

ANSWER: Change was done, Page 2 line 50

Page 2, Line 67

…contribute to attracting their natural enemies (predators….

ANSWER: Change was done, Page 3 line 79

Page 2, Lin 69

Parasitoids characteristics such as their………..

ANSWER: Change was done, Page 3 line 81

Page 2, Line 73

………plants [32,33,38-41]. Author/s names missing? reported that the parasitoids…….

ANSWER: Change was done, Page 3 line 85

Starting Page 2, last paragraph lines 89 to 103, are a different (sans serif) typeface.

ANSWER: Change was done, Page 3, lines 101-109, font was changed

 Results

Page 3, Line 117

………twice as much, on average, then than……..

ANSWER: Change was done, Page 4 line 131

Page 4, Figure 2. Axis labels - Frecuency should be Frequency– as correctly stated in the Figure legend.

ANSWER: Change was done, Page 6

Discussion

Page 7, Paragraph 2, lines 203 to 207, are a different (sans serif) typeface.

ANSWER: Change was done, Page 10, lines 220-224, font was changed

Page 7, Line 213

….documented in locations with very particular environments distinctive environmental conditions [68].

ANSWER: Changes were done, Page 10, lines 230 and 231

Page 7, Line 244

…under natural conditions (with ants ’s), which could be due to the……

ANSWER: Change was done, Page 11, line 263

Page 7, Lines 276 to 284 are a different (sans serif) typeface.

ANSWER: Change was done, Page 12, lines 270-281, font was changed

Page 7, Line 288

but they provide pay substantial attention to the scales on…….

ANSWER: Change was done, Page 13, line 306

Page 7, Line 300

little collateral direct benefit (or advantage)…..

ANSWER: Change was done, Page 13, line 319

Methods

Page 9, Line 312

….phytophagous hemipteran species, known as scales…..

ANSWER: Change was done, Page 13, line 332

Acknowledgements are also in different typeface and large point size.

ANSWER: Change was done, Page 16, lines 426-431

Reviewer 2 Report

Manuscript ID: plants-2061643 entitled “Herbivory in Myrtillocactus geometrizans (Cactacea): Are Parasitoids an Indirect Defense or Collateral Benefit?” by Alicia Callejas et al. submitted to section: Plant Ecology is a well written manuscript that flows well from start to end. I am suggesting some minor revision before consideration in Plants MDPI, Special Issue: Evolution of Plant Defense to Herbivores.  Attached PDF has my notes in blue for authors’ revision and my main notes are:

·         Use full name (plants and Insects) with author, order, family when first mentioned in text.

·         Reader usually wants to read how the authors conducted the study then reads the results. Material and methods section should be before results, also change figure numbers accordingly. Unless this is the style for section: Plant Ecology.

·         Add more details on site of study with information on weather condition during this evaluation.

Author Response

We have meticulously reviewed the manuscript and have given answers to the comments stressed by you. In this cover letter we are presenting the point-by-point answers.

Review 2

Yo suggested some minor revision before consideration in Plants MDPI, Special Issue: Evolution of Plant Defense to Herbivores.  Attached PDF has my notes in blue for authors’ revision”

ANSWER FOR PDF NOTES

At the suggestion of reviewer 1, Title: Modified, now is “Herbivory in Myrtillocactus geometrizans (Cactaceae): Do Parasitoids provide Indirect Defense or a Direct Advantage?”

Summary: The word “parasitoidism” was changed to “parasitism

Introduction: Page 3, line 85, reference was added

Page 3, line 87, “sp” is not in italic

Page 3, line 87, species name was added Hylocereus undatus

Page 3, lines 89-100, font was changed

Results: Page 7, line 170, parentheses were removed

Page 8, line 180, The word “parasite” was changed to “parasitized” in text and graphic

Page 9, line 187, The word “parasite” was changed to “parasitized” in text and graphic

Discussion: Page 10, line 205, a comma was added

Page 10, line 212, genus was abbreviated

Page 10, lines 225-232, font was changed

Page 11, line 250, was modified in order to clarify sentence

Page 12, lines 276 and 278, The words “parasitoids and parasitoidism” were changed to “parasitism

Page 12, line 288, a reference was added

Page 12, lines 295-302, font was changed

Page 13, line 316, The word “parasitoidism” was changed to “parasitism

Methods: Page 14, line 350, “sp” is not in italic

Page 14, In Figure 6 some size bars were added

Page 14, lines 371-376, GPS location and some general environmental conditions were added

Page 16, line 411, the sampled year was added

Page 16, lines 431-435, font was changed

Page 16 , lines 427, Person mentioned in this paragraph doesn't have some institutional affiliation at this moment

References: Pages 16-24, All references have been checked and modified where was necessary
